# Modelling Exposure by Spraying Activities—Status and Future Needs

**DOI:** 10.3390/ijerph18157737

**Published:** 2021-07-21

**Authors:** Stefan Hahn, Jessica Meyer, Michael Roitzsch, Christiaan Delmaar, Wolfgang Koch, Janine Schwarz, Astrid Heiland, Thomas Schendel, Christian Jung, Urs Schlüter

**Affiliations:** 1Fraunhofer Institute for Toxicology and Experimental Medicine ITEM, Nikolai-Fuchs-Straße 1, 30625 Hannover, Germany; wolfgang.koch@item.fraunhofer.de; 2Federal Institute for Occupational Safety and Health BAuA, Friedrich-Henkel-Weg 1-25, 44149 Dortmund, Germany; Meyer.Jessica@baua.bund.de (J.M.); roitzsch.michael@baua.bund.de (M.R.); Schwarz.Janine@baua.bund.de (J.S.); schlueter.urs@baua.bund.de (U.S.); 3National Institute for Public Health and the Environment RIVM, PB 1, 3720 Bilthoven, The Netherlands; christiaan.delmaar@rivm.nl; 4Federal Institute for Risk Assessment BfR, Max-Dohrn-Straße 8–10, 10589 Berlin, Germany; Astrid.Heiland@bfr.bund.de (A.H.); thomas.schendel@bfr.bund.de (T.S.); christian.jung@bfr.bund.de (C.J.)

**Keywords:** spraying, exposure models, occupational exposure, consumer exposure, evaluation, regulatory chemistry

## Abstract

Spray applications enable a uniform distribution of substances on surfaces in a highly efficient manner, and thus can be found at workplaces as well as in consumer environments. A systematic literature review on modelling exposure by spraying activities has been conducted and status and further needs have been discussed with experts at a symposium. This review summarizes the current knowledge about models and their level of conservatism and accuracy. We found that extraction of relevant information on model performance for spraying from published studies and interpretation of model accuracy proved to be challenging, as the studies often accounted for only a small part of potential spray applications. To achieve a better quality of exposure estimates in the future, more systematic evaluation of models is beneficial, taking into account a representative variety of spray equipment and application patterns. Model predictions could be improved by more accurate consideration of variation in spray equipment. Inter-model harmonization with regard to spray input parameters and appropriate grouping of spray exposure situations is recommended. From a user perspective, a platform or database with information on different spraying equipment and techniques and agreed standard parameters for specific spraying scenarios from different regulations may be useful.

## 1. Introduction

Spraying is an activity that leads to exposure to products and chemicals for workers and consumers. It leads to increased inhalation exposure to aerosols and thus spraying has already played an early role in the risk assessment for chemicals [1,2]. Substances as components of spray products are relevant for occupational as well as consumer risk assessment in regulatory contexts such as REACH (Regulation (EC) No 1907/2006), plant protection products regulation (“PPPR”, Regulation (EC) No 1107/2009) or biocidal products regulation (“BPR”, Regulation (EU) No 528/2012). As part of the risk assessment, human exposure during spraying activities primarily focuses on inhalation exposure to aerosols and vapours. For dermal exposure, deposition of aerosol and vapours is regarded as being important [3,4].

Spraying is performed in both occupational and consumer settings. The used spray equipment, work patterns, and resulting exposure patterns can be quite diverse [5,6]. Occupational applications include paint spraying [7,8], pesticide spraying [9,10], spraying of wood preservatives [11], disinfection processes [12], and industrial applications such as paint booth spraying or aseptic packaging [13]. Spray applications are common for a wide range of consumer products, e.g., household products for air freshening, cleaning, waterproofing, cosmetics as well as for do-it-yourself (DIY) activities. Common spray equipment includes aerosol spray cans, trigger sprayer, electrical devices (airless sprayers, high volume low pressure (HVLP) sprayers), hand-held pressure pump sprayers, compression sprayers, knapsack sprayers, or powered spraying devices such as tractor mounted sprayers.

The inhalation exposure to sprayed material originates from two sources: (1) primary exposure from the spray cloud and its overspray (see Figure 1); (2) secondary exposure caused by the evaporation of substances from the treated surfaces.

The exposure concentration is determined by the source strength, i.e., by the mass generation rate of airborne particles, the particle size distribution, and the dispersion of the released material in the room air. The source strength can vary by several orders of magnitude [14,15], depending on the spraying mechanism and technique (e.g., equipment, pressure, etc.) as well as on the composition of the spray formulation and its physical and chemical properties, such as vapour pressure. Aerosol formation and exposure during spray applications are the subject of ongoing scientific studies [5,9,15,16]. In the broader sense, spraying also includes activities such as foaming or fogging, foaming results in aerosol and vapour release as well [16]. For systematic exposure assessments, spraying can usually be divided into two sub-categories, space spraying and surface spraying. These can then be differentiated further by taking into account spraying equipment and technique, level of automation and containment, application rate, or direction of spraying. Details of exposure modelling depend on the target quantity to be evaluated, i.e., non-volatile substances dissolved in a solvent or the volatile solvent itself. For aerosol dispersion, parameters such as ventilation rate and turbulent mixing as well as droplet settling and droplet maturation due to solvent evaporation [17,18] have to be considered.

Dermal exposure during spraying activities is caused, among others, by aerosol and vapour deposition on the human body surface [4,19] resulting in a relatively homogeneous distribution of the spray liquid across the human body. Additionally dermal exposures are the result of accidental splashes of the liquid formulation during preparation and direct contact with contaminated surfaces. McNally et al. [3] found that for hand exposure, the deposition route was only important for spray activities and not so much for other activities. The authors reported that hand exposure may in addition be driven by contact transfer or direct emission and direct contact for these scenarios.

Different exposure models and tools exist. They contain specific approaches to estimate exposure to sprays. Lists of models are given in the BPR and REACH guidance [20,21,22]. In recent years, the performance has been validated or at least evaluated for a number of exposure models that predict human exposure for spraying [6,12,23,24,25,26]. The evaluation of spray models is complicated due to the high diversity in spray equipment and application patterns. The assessment of specific substances and products for regulatory purposes requires models that allow a more ‘general’ view of the application because they have to cover a wide range of situations. Health and safety regulations require that the variability in the parameters occurring in reality should be conservative. A conservative estimation of several model input parameters may lead to an unrealistic high exposure estimate. If sufficient information is available, variability could also be expressed quantitatively, for example, by handling ranges, or performing probabilistic evaluations. For Occupational Safety and Health (OSH), the evaluation of a specific situation at a specific workplace demands a more accurate estimation, offering maximum flexibility to simulate specific scenarios. Remaining uncertainties should likewise be dealt with by conservative assumptions, but in contrast to the above-mentioned regulatory assessments, the range of applicable parameters is usually much smaller. Considering both assessment goals, the implementation of a tool becomes more of a balancing act, in which addressing the recommendations for either goal may conflict. For this reason, the user of a model requires information on the applicability domain and/or uncertainties of each model.

Separate from the evaluation of the existing models, the scientific work in recent years focused on exposure studies for specific spray applications, equipment, and relevant process parameters. Großkopf et al. [9,27] developed new exposure models for outdoor and greenhouse pesticide applications for inhalation and dermal exposure based on a comprehensive set of measurements at workplaces across Europe. Schwarz et al. [16] studied the potential of aerosol release during spraying and foaming activities using different devices. Schwarz and Koch [15] presented a simple mass balance method for the measurement of the release fraction of thoracic and respirable particles of non-volatile compounds. Roitzsch et al. [5] studied dermal and inhalation exposure of certified pest control operators resulting from spraying with vehicle-mounted and handheld spraying devices. Kim et al. [28] characterized the behaviour of airborne particles generated from four categories of consumer spray products, and Laycock et al. [29] studied consumer sprays containing nano particles.

Although a certain understanding of exposure during spraying activities exist, and some models are available to predict exposure concentrations, there are still some deficiencies, especially about the applicability and uncertainty of the models. Given the continued importance of spraying activities for human exposure, the aim of this publication is to identify the needs for further development of respective exposure models. We concentrate on exposure towards liquid sprays and spray foams. For this, the following approach has been applied: (a) a literature search was performed to gather information on potential models and tools for assessment of human exposure during spraying activities (b) an overview of the existing evaluation of model performance was created and (c) possible influencing factors were discussed and needs derived.

## 2. Literature Review

To gather all relevant information, two approaches were applied. In a first step, a workshop was organized to get information from experts about already existing tools, the currents status, and the needs.

In a second step, a systematic literature search and review was performed, including information about tools and models accepted within different European regulations, the background documents, and about the evaluation status of the tools and models.

### 2.1. Workshop

The current status and further needs for modelling exposure of spraying activities were discussed at a symposium (WE-SY-B3) during the ISES-ISIAQ Conference 2019 [30]. The aim of the symposium was to offer a platform for a balanced discussion with input from science, authorities, and stakeholders. New scientific information and knowledge on modelling of spraying activities were presented and discussed. The symposium started with an introduction to the topic by S. Hahn, followed by a presentation highlighting “basic relevant parameters in the modelling of spraying activities” by W. Koch, and two contributions by J. Meyer and C. Jung, describing experiences, challenges, and desired improvements for workplace and consumer spray exposure assessment.

### 2.2. Systematic Literature Search

A systematic literature search on the available information on model and tools for estimating human exposure by spraying activities has been performed in Pubmed, WebOfScience, and Scopus.

Search term 1: exposure AND spraying AND model AND (occupational OR work* OR consumer)Search term 2: exposure AND (model OR tool) AND (robustness OR validation).

This search indicated 73 papers assessed for eligibility, from which 41 have been used for this review. The search was complemented by screening other sources such as websites of organizations (ECHA (European Chemicals Agency), EFSA (European Food Safety Authority), U.S. EPA (United States Environmental Protection Agency), OECD (Organisation for Economic Co-operation and Development), and WHO (World Health Organization)) and human exposure models. This identification resulted in 12 documents from organizational websites, and 34 documents from the model or project websites. This has been completed by 58 documents from expert knowledge of the authors, and 38 key publication from secondary citations. The search terms and the evaluation is presented in the Appendix A and illustrated in a PRISMA flow diagram.

Overall, after eliminating duplicates, this resulted in a total of 146 papers and documents which were cited in the present review. Thereof, 31 documents have been used for the description of models and tools (see Section 4), and 33 documents have been identified, which contain information about the performance of the models (see Section 5).

## 3. Categorization and Grouping of Models

A comprehensive overview on models and tools intended to estimate human exposure to spray mists will be given in Section 4. The terms “model” and “tool” are often used synonymously in the context of exposure modelling, although they have a slightly different meaning. While “model” describes a theoretical concept and any data records stored therein, “tool” means a specific implementation of one (or more) models. For example, ConsExpo (RIVM, Bilthoven, Netherlands) is a tool in which several models are implemented. Often a tool is the implementation of a model with the same name, but not necessarily the only one. For example, the Advanced REACH Tool (ART) webpage hosts an implementation of the model “ART” [31,32], the tool TREXMO contains another implementation of this model [33,34], and in principle it is possible to perform an estimation with the model ART, following its published descriptions without using either of these tools. We will here focus on the models, their scientific base, and the parameters used for modelling exposure scenarios.

Exposure models can be categorized according to different schemes. We based our analysis on the WHO International Programme on Chemical Safety [35], which has suggested a classification scheme based on two criteria.

The first criterion relates to the internal conception of the model. Mechanistic (also physical) models aim to model the exposure based on the simulation of basic physical equations (typically mass balance differential equations). Empirical models are based on the statistical analysis of measured exposure levels. For an exact definition, see [35].

The second criterion refers to the way in which the exposure estimate is output by the models. Deterministic models, for example, generate a discrete result based on a set of (discrete) input parameters. On the other hand, stochastic or probabilistic models will take into account the natural variability of an exposure situation and the uncertainty in the model inputs and will estimate a distribution of the resulting exposure. Some models provide percentiles as an output which can be considered as stochastic although often only one (deterministic) value is recommended for further use.

In addition to the criteria of this classification scheme, an additional criterion could be assigned to the type of the input parameters. In some models, numerical values, e.g., a vapour pressure, are required as input parameters, while other models require the selection of categories, e.g., fugacity bands. Most models use a mixture of these parameter types.

Exposure models often combine some of these aspects making them hybrid models.

Finally the scope of models may differ. Some are only intended for screening purposes (often termed “lower tier” or “tier-1” models), i.e., to pre-select the more critical scenarios based on a very limited set of input parameters, while others are designed to reflect the exposure situation more precisely (“higher tier or “tier-2” models). This is important, as the first type of models should tend to overestimate, whereas the second should estimate closer to the actually occurring exposure. The scope of the models will therefore have an impact on the evaluation of their performance and their ability to represent precisely the real extent of exposure.

## 4. Description of Existing Models

In the following, we will give brief descriptions of some common exposure models capable of modelling exposure to sprays. A short description of the underlying model structure as well as the input parameters and the output data are given in Appendix A. Models that are no longer publically available as tool and/or are no longer used in the European chemical regulations as well as models in a very early development or demonstration status such as models described by Brouwer et al. [36] or Semple et al. [37] were excluded from further analysis (Appendix A). Likewise, models that were developed specifically for the nanoscale, models that do not calculate personal chemical exposure or estimate secondary exposures, e.g., due to spray drift in outdoor, applications or do not calculate quantitative exposure were excluded from the present analysis. Table 1 provides an initial overview of models and tools recommended and usually used for exposure estimation of spraying activities, especially under EU legislations.

The “TNsG Database Detailed Models” [20,61,62] is an example of a collection of simple empirical exposure models used for biocides in the EU. A major advantage of these models is the limited amount of information required to generate an exposure estimate. The TNsG contains amongst others 14 datasets that have been measured for professional spraying or fogging processes, and three datasets for related consumer spraying applications. Separate exposure statistics for each of the different datasets are available. The datasets contain measurements from the same industry (e.g., disinfection in the meat processing industry, or spraying of insecticides) and/or it represents comparable spray equipment and/or situations such as direction of spraying or pressure. 

Some empirical tools, such as RISKOFDERM [56,57] or the Agricultural Operator Exposure Model (AOEM) [9,27,40] employ elaborated statistical approaches such as regression analysis performed on the exposure data during model development. It should be noted that the datasets used to develop empirical models are also often used to develop more than one model. For instance, some of the data used for the development of RISKOFDERM or the TNsG models have also been used in the Bayesian Exposure Assessment Tool (BEAT).

RISKOFDERM was developed and designed to estimate dermal exposure values. RISKOFDERM separates processes into Dermal Exposure Operation units (DEO units). Spraying activities are addressed in DEO unit 4 where specific parameters such as spray direction, segregation from source, application rate, duration, and other information are addressed. The model supplies exposure estimates for both, hands and body separately.

The AOEM estimates dermal and inhalation exposure resulting from treatment of high or low crops with tractor mounted and handheld spray equipment performed outdoors and from treatment with handheld spray equipment in greenhouses. It is based on empirical data from measurement campaigns performed between 1994 and 2009. The reports cited here include comprehensive descriptions of all included datasets. The AOEM is currently the standard model for assessment of agricultural operator exposure recommended by the EFSA [63].

The next level of abstraction comes with models such as ECETOC TRA for worker (ECETOC, Brussel, Belgium) or MEASE (EBRC, Hannover, Germany). These models were developed by analyzing exposure measurements at the workplace. However, the models incorporates expert knowledge and judgement, and simple mechanistic approaches.

The ECETOC Targeted Risk Assessment (TRA) tool for worker [48] is a MS Excel based exposure estimation tool mainly used in registrations under REACH. It is a source-receptor model generated from measured data, which is based on the process categories (PROCs) of the REACH descriptor system with regard to the mode of use. For all different existing spraying activities at workplaces, only two process categories are available, PROC 7 for industrial spraying and PROC 11 for non-industrial (professional) spraying. However, ECETOC TRA for workers will not capture the exposure to aerosol mists. For dermal exposure, ECETOC TRA for workers provides exposure estimates for the hands or forearms only. MEASE has partly the same model basis as the ECETOC TRA worker tool, it is also PROC-based and works with modifiers.

Other models such as Stoffenmanager^®^ (Cosanta B.V., Schiphol-Oost, Netherlands), the Advanced REACH Tool (ART) [31,32], as well as the dermal Advanced Reach Tool (dART) [3,46,47] or the BROWSE model [41,42,43] might best be considered hybrid models. A conceptual framework based on physico-chemical laws as well as measurements in combination with elements of expert judgement give these models mechanistic and empirical properties. Stoffenmanager^®^ and ART, for example, make use of modifying factors and calibration factors derived from measured exposure levels in order to generate an exposure estimate. Stoffenmanager^®^ is a tool to assist small and medium sized enterprises with chemical risk assessments of workers in the framework of OSH and national legislations. Additionally, the tool offers a quantitative inhalation exposure module. It considers aspects of the product (solid, liquid), process (type of task, duration and frequency, distance to the task, protection of the worker), and workplace (description of the working room, worker situation, protection of the worker). The measurement data in Stoffenmanager^®^ used for calibration stem from an underlying non-public database in which numerous published and unpublished workplace measurements were collected. ART was developed for inhalation exposure assessments under REACH. It considers the substance and activity emission potential, localized controls, segregation, separation, worker behaviour, surface contamination, and dispersion. The ART model allows to upload additional exposure data in order to carry out a Bayesian update of the mechanistic model prediction. It also uses a probabilistic approach based on the modelled median value and random numbers to estimate distributions of the expected exposure levels.

The refinement options offered by these more generic models promise assessments of a wider range of spray scenarios in comparison to the purely empirical models. Especially with generic models such as ECETOC TRA for workers or ART, by which highly abstract exposure situations, the choice of parameters by the user, and the data treatment performed by the algorithms may introduce uncertainties in the estimated exposure levels [64]. On the other hand, these models may also miss input determinants, especially for spraying activities.

Simple mechanistic models such as the ECETOC Targeted Risk Assessment (TRA) tool for consumers (ECETOC, Brussel, Belgium) or the A.I.S.E. REACH Exposure Assessment Consumer Tool (REACT) (International Association for Soaps, Detergents and Maintenance Products, Brussel, Belgium) as well as the “instantaneous release” model of ConsExpo assume as a worst-case approach that the complete amount sprayed is available for inhalation. EGRET (European Solvents Industry Group, Brussel, Belgium) is an adaptation of the ECETOC TRA tool for consumers targeted at exposure to solvents. These models also feature conservative dermal exposure estimation methods. For example, assuming that the entire product is applied directly to the skin instantaneously or at a constant rate. The more advanced Consumer Exposure Model (CEM) of the U.S. EPA offers, amongst others, to estimate exposure due to spraying activities. The scenario considers inhalation of the overspray, as well as exposure to substances after evaporation from a treated surface. The exposure can be estimated for the breathing zone of the user or alternatively for a homogenous well-mixed room. The model allows further to distinguish between time periods where the product is used and when it is not.

The spray exposure models implemented in ConsExpo (RIVM, Bilthoven, Netherlands) and SprayExpo (BAuA, Dortmund, Germany), represent more sophisticated mechanistic approaches. They integrate physics-based mass-balance equations to simulate the time development of the air concentration and inhalation exposure to the aerosol generated by the spray process. Source terms in these models describe the emission of aerosol. Sink terms include gravitational settling and ventilation. The aerosol formation characteristics and the equipment specific characteristics are typically not well known and require considerable effort to measure. For both models, the particle diameter of the generated aerosol is an important exposure factor. It determines the calculated settling velocity as well as the deposition of aerosol in the respiratory tract. The latter is considered in a simplified way in ConsExpo by means of a cut-off diameter below which particles are assumed to deposit in the alveolar region [45]. Regarding the model output both, ConsExpo and SprayExpo are mostly used as deterministic tools; however, ConsExpo does also accept ranges of input parameters and uses them to calculate distributions of exposures employing a probabilistic approach.

## 5. Evaluation of the Performance of the Models

To get reliable exposure estimates, model validation is necessary to evaluate their performance. However, exposure situations and circumstances vary greatly between scenarios and persons. It is practically impossible to tightly specify the exposure conditions [65] and conduct a complete set of experiments that adequately covers all conceivable exposure situations that may be addressed with models and tools. Therefore, we prefer here to refer to the experimental testing of exposure models as ‘evaluation’ in line with Oreskes (1998) [66]. 

For Tischer et al. [67,68], the evaluation of a model includes a conceptual evaluation and uncertainty analysis, an operational analysis and an external validation.

Conceptual evaluation and uncertainty analysis is a review of the implemented methods and assumptions, ascertaining that the algorithms used are scientifically plausible and their implementation is correct [68]. Uncertainty in the available information and the exposure assessment method are dealt with by using conservative assumptions.

Operational analysis contains aspects on user-friendliness and ‘between-user reliability´ of the tool [68]. User-friendliness includes evaluation of guidance, documentation, and transparency of the tool for model users. Between-user reliability considers that the exposure estimate is reproducible and robust in a sense that conclusions on safety will not vary from one assessor to the next. 

External evaluation will typically test partial or intermediate predictions of a model against measurable quantities in a limited set of well-controllable experimental scenarios and is therefore, strictly speaking, limited to the considered domain of exposure scenarios. 

Therefore, a model evaluation will not prove a model to be false or correct, but will test its plausibility and in this way build confidence in the use of a model. Model evaluation is essential to support and justify the use of exposure models in regulatory exposure assessment.

It has to be noted, that the models are often developed for a specific scope but this applicability domain is often extended in reality. For example, ConsExpo was developed to assess consumer exposure, but is also used by exposure assessors for occupational settings using adapted input parameters. Advanced REACH Tool (ART) [31,32] is used not only for workplaces under REACH (Registration, Evaluation, Authorisation and Restriction of Chemicals) for which it was developed but has been used under other regulations such as BPR as well.

To summarize, the reliability of exposure assessments made using exposure models is the combined result of the precision and accuracy of the exposure model underlying the tool, its implementation in the tool, and the use of the tool including the quality of the information base used as input of an assessment [68].

In the following, published evaluation studies of exposure assessments for spraying activities at the workplace or for consumer use are summarized. Appendix A gives a detailed overview on the studies which include spraying processes as part of an evaluation. However, spraying is mostly considered only as one of several activities.

### 5.1. Information on External Evaluation of the Models—Inhalation Exposure

Data-based models such as the TNsG models are based on measurements for specific situations or a set of measurements with similar application or equipment. While these models reflect the exposure occurring in the source scenario very well, they are in principle only accurate and thus reliable for a small range of scenarios where substances with similar properties are examined under similar use situations as in the datasets (e.g., vapour pressure, duration, direction of spraying, spraying equipment, pressure, room volume, etc.). They may be useful for a wider range of situations as a worst-case estimate at least for screening purposes if they can clearly overestimate the exposure of the target situation. However, an exact description of the situations covered in the model’s dataset is often lacking; thus, the validity domain boundaries and the accuracy and reliability for the target situation cannot always be well assessed. In limited cases, for instance, the study by Roitzsch et al. [5], specific comparisons are available but the discrepancies found there can be interpreted by actual differences between the investigated situations. Overall, it would be favourable if a more exact description of the measured situations are available and the range of situations covered by the specific models would be given. The ongoing CEFIC LRI project [69] on “Extrapolating the applicability of worker exposure measurement data” studies in a broader sense this extrapolating issue and the conditions and uncertainties.

The reliability of the models ART, Stoffenmanager^®^, and ECETOC TRA for workers were assessed in a number of studies, most of which are summarized in Spinazzè et al. [70], Lee and Lee [71,72]. For evaluation of our literature search for the models ART, Stoffenmanager^®^, and ECETOC TRA for workers, we focused on evaluation studies that are based on previous workplace measurement data and/or on measurement data collected at workplaces explicitly for the purpose of the studies (see Appendix A). Except for Hofstetter et al. [73], where measurements were gained under laboratory conditions and not at real workplaces. Most studies did not specifically focus on spraying. Spray activities in the studies included, but were not limited to pesticide spraying, spray painting, spraying of plastics, gel coating by spraying, and cleaning by spraying. With regard to the assignment of activities to PROCs, we noticed that spray painting was assigned to both PROC 7 and PROC 11 and thus seems to be relevant for industrial and non-industrial settings. 

For ECETOC TRA (worker model, v2 and v3), a total of eight studies (see Appendix A) were evaluated that involve different spraying activities. Landberg et al. [74] pointed out that, according to ECHA, ECETOC TRA for workers is not suitable for the estimation of exposure to aerosols (aerosol mists). For industrial spraying (PROC 7), the general tenor of most studies taking spray activities into account is that ECETOC TRA for workers does not provide sufficiently conservative exposure estimates. For non-industrial spraying (PROC 11), on the other hand, ECETOC TRA for workers was found to provide conservative estimates. The reported levels of conservatism for PROC 11 span between sufficient and very conservative, e.g., as reported by Lamb et al. [75] or Vink et al. [76]. As Spinazzè et al. [77] found unrealistically high exposure estimates for pesticide spray activities, they suggest not to use ECETOC TRA for workers for exposure estimation of spraying of diluted chemicals with very low volatility. Furthermore, it was a general result of the ETEAM study [24,75] that version 3 of ECETOC TRA for workers estimates the exposure less conservatively than version 2.

For Stoffenmanager^®^, 11 studies were evaluated (see Appendix A) in which spraying activities were included. A main conclusion of most studies is that Stoffenmanager^®^ is a balanced model with a medium level of conservatism, when considering the 90th percentile [58,78,79]. The few studies that examined the 50th percentile came to a similar conclusion. According to Spinazzè et al. [77], Stoffenmanager^®^ is the most robust tool against uncertainties in model input. The authors recommend using the tool in cases where the model input information is uncertain or difficult to interpret. One notable result of the ETEAM study was that Stoffenmanager^®^ is not sufficiently conservative for non-volatile liquids [75] in PROC 11.

For ART, nine studies were evaluated (see Appendix A) that compare the estimates with measurement data including spray activities. The studies mostly focused on the 50th or 90th percentiles. Thereby, the 90th percentile represents a conservative estimate and the 50th percentile a "reasonably accurate" estimate of the exposure. A few studies even considered the 90% confidence interval (CI) of the 90th percentile. Spinazzè et al. [77] reported that the 90th percentile leads to an underestimation for spraying of pesticides. Most other studies reported conservative estimates, although the degree of conservatism specifically for spraying activities is difficult to extract from these studies. According to Spinazzè et al. [77], ART may best be used when the exposure scenarios to be assessed are well-documented. And Lee et al. [72] suggested to use the 90% confidence interval of the 90th percentile. They found that the median prediction appeared accurate, whereas using the 90th percentile 41% of the measurements were still above the model results. The authors suggest re-examining the assumed variance in the model, as there was an unexpected underestimation of the 90th percentile by the model. The above statements refer to vapours. The sample size for spray activities in this study was small and no separate statistical analysis was conducted. However, the study seems to indicate similar trends for spray activities.

Simple mechanistic approaches to simulate spraying activities such as ECETOC TRA for consumers or AISE REACT assume that the substances evaporate immediately (instantaneous release). These approaches aim at conservatism and in theory will overestimate true exposure levels. Regarding the consumer version 3 of ECETOC TRA, Oltmanns et al. [80] contrasted model results for several scenarios of this model and similar low-tier models like EGRET and AISE REACT with the higher-tier models of ConsExpo 4.1. This comparison also contained two spray applications with volatile compounds (paint spray can and glass cleaner trigger spray). The authors showed that the exposure results were generally higher than for ConsExpo 4.1; therefore, the conservative aim of these rather simple models seemed plausible for the examples used. Regarding AISE REACT, to the best of our knowledge, no evaluation study exists.

Mechanistic models such as SprayExpo and ConsExpo relate emission source characteristics to exposure by quantitatively describing the physical processes of translocation and removal of aerosols from the air. The well-established and tested scientific basis of these models makes them reliable in theory. However, in practice, such models need to balance model complexity with information availability in order to allow practical operation of the model, which often leads to a simplified, less realistic model architecture. For example, ConsExpo does not consider the physics of evaporation of spray components from the aerosol droplets after release, as accurate modelling of evaporation requires detailed information on product composition, which in practice is seldom available.

A number of evaluations of the mechanistic models SprayExpo and ConsExpo have been performed in the last couple of decades. Eickman et al. [81] conducted an extensive review of these two (and some other) tools. A sensitivity analysis was done to identify the relevance of different model assumptions. The authors also discussed the usability aspects of the tools. The predictions of the models were tested against measured air concentrations from 10 different occupational spray applications. The authors found that the SprayExpo model simulations were overall within a factor of 5 from the measured values. By contrast, the results of the ConsExpo 4.1 simulation underestimated the measured data by several orders of magnitude. The authors attributed this to the fact that ConsExpo did not include evaporation of solvents or active ingredients, which was expected to be significant for the studied products.

Beyond that, the SprayExpo tool has been evaluated by dedicated spray experiments [6,14]. A sensitivity analysis revealed that the (specification of the) aerosol particle size distribution has a critical impact on the predicted exposure levels. The study determined air concentrations arising after application of 11 biocidal spray, fogging, and nebulizing products for the protection of stored goods. Different size fractions of the aerosol, representing different deposition classes in the human respiratory tract were collected up to 30 min after spraying. In this study SprayExpo tended to predict more accurately the experimental air concentration levels, whereas ConsExpo tended to overestimate (somewhat) the concentrations for air space applications. However, the degree of overestimation by ConsExpo for surface sprays was even more pronounced. The authors attributed the overestimation by ConsExpo to the model simplifications made for the volatilization of the spray components.

For the special case of applying the spray exposure models to nanomaterials in sprays, Park et al. [82] conducted experiments on the exposure to nano-silver released from an indoor air deodorizer. The experimentally measured inhaled dose values were compared with model simulations of SprayExpo, U.S. EPA’s CEM and the spraying model in ConsExpo. Experimentation was done in two extreme ventilation regimes: one in the absence of any ventilation, the other at an unrepresentatively high value of ventilation of 35 air changes per hour. The authors concluded that in these cases, the model predictions of the tested tools tended to describe the short-term experimental exposures reasonably well, but the models deviated more strongly from the experimental measurements at longer exposure times. However, some ambiguity about these results exists: the authors claim that for one experimental setup, the inhaled dose is actually smaller for longer exposure times. From a mass balance point of view, this seems impossible.

Regarding biocidal spray products, Clausen et al. [12] performed spray experiments and compared them with simulation results from the spraying model in ConsExpo. Three biocidal sprays (two insecticides and one disinfectant) were tested in a climate test chamber with an air exchange rate of 0.5/h. The released particles, airborne organic compounds in the gas and particle phase, and the surface concentrations of the organic compounds were measured. The particle size distributions were determined after 5–10 min. As input for the spraying model of ConsExpo, lognormal distributions were estimated using the geometric mean and standard deviation of the experimentally derived distributions. The authors concluded that the peak concentration is underestimated by at least a factor of two but that gravitational settling is also underestimated in the model, since the decline of the air concentration appeared slower in their model simulation, compared with experimental observations.

Delmaar and Meesters [26] conducted an evaluation of ConsExpo’s spraying model by testing the model’s prediction with experimental measurements from multiple studies on consumer products published in literature. In total, 19 different data sets from five studies were identified that were deemed suitable for model evaluation. One observation made by the authors when screening potentially useful studies was that in many instances, the information on the study was insufficient for an accurate model evaluation. Critical information that was often lacking in study descriptions was information on product composition, mass of product used, and aerosol that the spray product generated. As even the selected studies did not provide conclusive descriptions of the experimental setup, the evaluation was performed, including a quantitative analysis of the uncertainty in the experimental specification. In most cases, the model results were compatible with the measurements, when considering the uncertainty in the experimental setups. Model predictions tended to be more inaccurate in the initial stages of exposure, shortly after application of the product. The fact that evaporation of ingredients from the spray was not accounted for could lead to additional inaccuracy, but its quantitative effect could not be determined with the data available. One particular observation made in the evaluation was that particle size distribution is a critical parameter for exposure estimation, whereas in many of the experiments, this aspect was only partly (and insufficiently) documented. The authors made recommendations on the setup and documentation of experimental studies for the evaluation of aerosol exposure models.

Recently Park and Lee [83] undertook an evaluation of the ConsExpo spray model, contrasting model simulations with both experimental concentrations after biocide spray application and with computational fluid dynamics (CFD) simulations. The experiment followed particle concentrations during 60 min in the air of a 30 m^3^ climate chamber. CFD model and experiment were in good agreement over the entire experiment duration. The ConsExpo simulations agreed very well with the far-field measurement and CFD simulations, but underestimated the time weighted average of the near-field concentrations by a factor of 5. The study only evaluated the ‘well-mixed’ room mode of the ConsExpo model and not the ‘near field’ (i.e., the ‘spraying towards person’) mode.

The studies on model evaluation with experimental data described above did not present a coherent picture. For example, Eickman et al. [81] found that the ConsExpo model predictions were orders of magnitude below that of the measured air concentrations. In contrast, Koch et al. [6] observed that ConsExpo tended to overpredict the experimental air concentrations in their setup. Additionally Delmaar and Meesters [26], as well as Park and Lee [83] observed largest deviations between ConsExpo’s model prediction and measurement in the early stages of exposure, whereas Park et al. [82] found that the evaluated tools all tended to deviate from the measurements in the later stages of the exposure experiment. The uncertainty of both input parameters and measurements could be one of the reasons for the observed inter-study differences. On the other hand, these differences in the conclusions of evaluations may reflect true inter-experimental variation (in particular, variation in selected products and studied substances) and the suitability of the models to capture the specific experimental setting correctly. However, it may also stem from differences in the use of the modelling tools and interpretation of the models and their input parameters among the researchers involved in the different studies. Comparing experimental results with model simulations is further complicated if not an actually measured particle size distribution, but only a fitted or even estimated one is used as input for the models, or if the actual maturation of the droplets is not accounted for, i.e., if the particle size distribution is measured a significant time after the spray event occurred.

### 5.2. Information on External Evaluation of the Models—Dermal Exposure

In general, the dermal models are less well evaluated than the inhalation models. Cresti et al. [84] considered the scenario of spraying a disinfectant on a hard surface and compared the results of several data based models with RISKOFDERM and ConsExpo 4.1. For the investigated scenario, the models indicate that the dermal exposure is higher compared to inhalation exposure. Except for ConsExpo 4.1 (dermal: instant application), all models resulted in the same order of magnitude. However, the study is limited, because the model outputs are not compared to measurement data. Marquart et al. [85] collected data of hand exposure that were published in scientific reports and peer-reviewed publications as well as data from industry partners to compare these to ECETOC-TRA hand exposure estimates. For spraying of liquid substances and solid substances in liquids in PROCs 7 and 11, the authors found that ECETOC TRA for workers tended to overestimate the 75th percentile of the measured hand exposures. Vink et al. [76] compared the dermal outputs of ECETOC TRA for workers and RISKOFDERM. However, since no dermal measurements were included, only modelled results have been compared. The general conclusion was that RISKOFDERM estimates for hands under worst-case assumptions are higher than the corresponding ECETOC TRA for worker estimates. McNally et al. [3] performed a calibration of the dART model with published data. For the spraying scenarios, the modelled and measured values used for calibration deviated by less than a factor of 10. At the OEESC Dublin 2019, Goede et al. [46] presented preliminary results of a comparison between dART, RISKOFDERM, and ECETOC TRA worker outputs for hand exposure with exposure data measured in test room situations during the SysDEA project [86]. The preliminary analysis indicated that values modelled with dART were higher compared to the SysDEA measurement data, followed by RISKOFDERM and ECETOC TRA for workers. Another preliminary result was that dART conservatively estimated hand exposure to low viscosity fluids, but slightly underestimated high viscosity fluids. Meyer et al. [87] made a comparison between SysDEA measurement data and RISKOFDERM model results for both hands and body. They found that for spray activities, the 75th and 90th percentile estimates of the RISKOFDERM model for hands and body were much higher than the corresponding measured exposure percentiles [87]. However, the results should be interpreted with caution, as the SysDEA project only examined controlled test room situations and not real workplaces.

### 5.3. Information on Operational Analysis of the Tools

Using telephone interviews and online questionnaires, Crawford et al. [88] assessed the usability and user-friendliness of the tools ECETOC TRA for workers (versions 2 and 3), MEASE v1.02.01, RISKOFDERM, and Stoffenmanager^®^ v4.5. In total, the response of 295 respondents from the industry, consultancy, government, research, and other organizations was taken into account. Overall, the documentation of the tools seemed to be satisfactory. The authors found that to some extent, the use of the tools seemed to be connected to the knowledge of the underlying tool concepts and the user’s experience. With regard to the spraying scenarios, the user feedback was that the assumptions made by MEASE, ECETOC TRA for workers, and Stoffenmanager^®^ seemed to be too abstract or too simplified to some users.

On the other hand, from personal experience of the authors, users often experience mechanistic models as too complex. A major problem is that these models have too many input parameters, of which the device-specific parameters (e.g., particle size distributions), especially, are often unknown.

Various studies such as Schinkel et al. [89], Landberg et al. [90], Lamb et al. [64], and Savic et al. [91] reported in the past that the exposure estimate for an exposure situation was subject to variations when a tool was used by different users. All studies reported amongst others that the choice of task class (process category (PROC) codes for ECETOC TRA for workers (v2 and v3) and MEASE, the task characterization input for Stoffenmanager^®^, the choice of activity/task for ART, and the Dermal Exposure Operations (DEO) in RISKOFDERM) had an important impact on the exposure outcome. Specifically for investigated spray scenarios, only Schinkel et al. [89] found that the assignment of the parameters spraying technique and application rate did not pose a problem for the users. Lamb et al. [64] suggested to implement additional support and quality control systems for all tool users. Schinkel et al. [89] proposed extensive training to improve the use of the tools.

## 6. Discussion, Derivation and Identification of Needs

Based on the workshop, the description of the existing models (Section 4) and the available evaluation of the performance of the models (Section 5) several needs for improving the modelling of exposure for spraying activities could be derived. The needs can be categorized for development and improvement of the models, for additional evaluation of the models, for sector and use specific information, and finally needs for modelling from a regulatory perspective.

### 6.1. Development and Improvement of Models

Several general (research) needs could be derived regarding the mechanistic knowledge and the influencing parameters. The existing models consider the exposure-determining parameters and the variability of the type and level of exposure in different ways. This is not surprising, as it must be noted that the models pursue different objectives, i.e., a conservative and/or a rather realistic estimation. 

Most of the models focus only on the estimation of the exposure occurring during the duration of the actual application. The models could be improved if additional exposure occurring during an additional stay or re-entry in the room after the spraying application will be covered.

Spray formulations consist often of non-volatile ingredients and (volatile) solvents. Additionally, the exposure to (semi-)volatile components of the spray can be of interest. Therefore, extension of mechanistic model approaches is needed to cover spraying of (semi-)volatile substances, i.e., combined exposure to spray mist, evaporation from droplets, and evaporation from treated surfaces. Evaporation of the (semi-)volatiles from the liquid phase, i.e., the vapour-droplet partitioning, depends on their effective equilibrium vapour pressures, which is determined by the chemical composition and thus the concentrations and activity coefficients of the constituents in the spray formulation. Currently there are no systematic studies on these aspects related to sprays. 

Further research is necessary to consider the type and amount of the variable source strength for different spray equipment in the modelling. Obviously the source strength for aerosol and/or vapour release of the applied spray technology is an important quantity determining the exposure of the user. In this context, it is useful to consider a source strength normalized to the total mass flow of the liquid (release fraction). Thereby model prediction could be more targeted by focusing on different particle size fractions (respirable, thoracic, inhalable). For example, Schwarz and Koch [15] found a simple correlation between thoracic source strength normalized to the total mass flux (release fraction) and the maturated median droplet diameter. Different types of spray equipment were investigated. The release fraction covered four orders of magnitude ranging from some ten percent for propellant sprays to 10^−2^ percent for liquid spray nozzles generating sprays with median droplet diameters above 100 µm. Therefore, a single scenario covering all spraying equipment is not adequate. For lower tier models, a categorization into different size distribution categories such as fine (propellant sprays), medium (air-assisted spray guns), and coarse (liquid spray nozzles) may be appropriate. The establishment of a database with source parameters for different spraying devices and techniques is beneficial, which can be used for spray model optimization and practical implementation of these concepts.

As information on the available models and their performance shows that foaming activities are not adequately covered, a need for development and evaluation of models for exposure assessment of foaming activities can be identified. Spray foams are used for example as disinfectant and cleaning foams as well as insecticides. It is supposed that foam applications have lower potential for inhalation exposure than comparable spray applications [16]. In contrast to droplet spraying, only very limited information exists on the generation of inhalable aerosols during foaming [16,92]. There is a need to identify and quantify the specific process parameters that control aerosol formation and to parameterize the release fraction with respect to the controlling parameters. So far, no model exists which specifically assesses foam activities but a model development is planned in the ongoing BAuA (Federal Institute for Occupational Safety and Health) project F2366 “Human exposure to biocidal products: Measurement of inhalation and dermal exposure during the application of biocide foams” [93].

The value of dermal exposure models for spraying activities are limited as well, and thus should be improved, developed and evaluated.

For example, relevant mass transfer processes for the different spraying activities and the main influencing parameters may need more detailed investigation for future dermal (hand and body) model development. Mechanistic modelling of dermal exposure can be coupled with mechanistic inhalation models. In these cases, dermal exposure is derived by relating the airborne concentration and the deposition flux on the human surface. However, this may not be the only relevant dermal exposure pathway, especially for the hands. The calibration of the dART model [3] indicated that the relative importance of the three mass transfer processes “aerosol deposition from the air”, “direct emission from splashes”, and “transfer through hand-to-surface contacts” is highly dependent on the specific spray situation. 

Body exposure should not be neglected for exposure assessments of spraying activities [5,87,94]. In this respect, an important limitation with regard to the dermal modelling approaches is that a number of models only provide exposure estimates for hand exposure and not for the entire body. 

In addition, dermal exposure due to spraying of (semi-) volatile substances is, to the best of our knowledge, not considered by modelling approaches or experimental investigations.

### 6.2. Needs for Additional Evaluation of the Models

As was pointed out in Section 5.1, the analyzed studies on model evaluation did not provide a coherent picture. Without a thorough meta-analysis of the different studies, it is not always possible to assess to what extent the studies are contradictory or consistent and which overarching conclusions can be drawn with respect to the conservatism of the models. Moreover, the review of existing evaluation studies revealed the need for a more thorough and targeted evaluation of the available models with a focus on spraying scenarios and taking into account the diversity of spray equipment.

In reality, repeated measurements of the same exposure situation will not lead to exactly the same exposure values. This is partly due to the fact that user behaviour influences the exposure level and that the conditions, e.g., at different workplaces, slightly differ. Empirical and hybrid models take this variance into account by presenting exposure estimates in form of percentiles. For deterministic and probabilistic models, an evaluation should ideally include a comparison of the different percentile or percentiles with the corresponding statistical values of the measured data.

Mechanistic models mimic the circumstances of the real world as accurately as possible through a rather complex model input and underlying model structure. The models therefore may output an exposure value that can be directly compared with individually measured exposure values.

For the hybrid models, ECETOC TRA, Stoffenmanager, and ART, we have shown in Appendix A when values deviating from the above concept were compared in the studies. These methodological limitations lead to a lower explanatory power with regard to evaluating the degree of conservatism of the examined models.

When evaluating the different models, it is also important to keep in mind the purpose or application of the models, i.e., whether they are intended for screening purposes or to reflect the exposure situation more accurately. The objective of the models therefore affects the degree of conservatism, which describes how much the model overestimates or underestimates the actual situation.

It would be desirable for the future to evaluate model performance on an activity-related basis and for a number of different percentiles of the model output. Spray equipment details as well as volatility of the investigated substances should also be explicitly described in these studies in order to be able to rank the usefulness of the models for the specific spraying scenario.

A few studies have been carried out to evaluate the mechanistic models in ConsExpo and SprayExpo with experimental data. Together, these studies could provide a reasonable database for evaluation purposes but the conclusions of the evaluation studies are not unambiguous and differ quite strongly in some aspects. The number of experimental studies is still lacking for a thorough evaluation of mechanistic spray models [26] and therefore further efforts in this field should be undertaken. Such experimental studies need a clear documentation of all the relevant parameters such as chemical composition, amount of product used, and detailed aerosol size distribution. For the latter, it is important to stress that (geometric) mean and standard deviation do not suffice [26]. In practice, it seems that most of the experimental studies from the public literature do not meet the criteria for sufficient documentation. Delmaar and Meesters [26] provide an overview of critical aspects that should be well described in an experiment to be useful for the evaluation of models. However, conducting additional experimental studies will not lead a priori to a more homogeneous insight into the performance of the different models.

Obviously, there is a need to confirm the findings reported in Section 5.2 by further evaluation studies on dermal exposure. The studies should take into account the diversity of spraying activities and equipment relevant for REACH and the models available to estimate body exposure. To the best of our knowledge, comparisons of body exposure measured at real workplaces and evaluation studies for consumer spray activities are completely missing. In general, studies comparing hand and body exposure measurements and model outputs are underrepresented although the few existing studies contain a comparatively large number of data points for spraying activities.

With regard to user friendliness and between-user reliability of the tools, the guidance and documentation of the tools could also be improved with respect to the spray equipment and spray tasks covered and training provided on how to use the tools.

### 6.3. Sector Specific and Use Specific Information

Next to increasing the knowledge about the mechanistic relations and influencing parameters, needs for information about user (worker, consumer) behaviour and the use scenario are identified to improve exposure assessment for spraying activities. The use scenario includes, for example, spray equipment and parameters such as use frequency, amount, application, and exposure duration. This information could be sector specific and use specific. The best would be to develop a database with agreed default parameters for specific spraying scenarios.

In the following existing sources of information is introduced and discussed, and which information would be additionally recommended to increase the reliability and decrease the uncertainty of the exposure assessment.

The available mechanistic models such as SprayExpo and the spraying model in ConsExpo usually focus on the exposure to the aerosolized non-volatile substance. Thereby, the droplet size distribution of the spray is an important input parameter determining the aerosol source term required, but the complete droplet size distribution function is not known for many equipment and situations. For this reason a need has been identified to compile this input parameter in dependence of equipment and situations. Some information is already available for specific equipment and articles in the ConsExpo reports and factsheets [23,95,96] and in the SprayExpo research reports [6]. This list of droplet size distribution could be a part of the database mentioned above.

Some information about consumer behaviour is available. Available studies on consumer behaviour mostly measure the frequency of spray applications [97]. Product amounts and application duration are surveyed to a lesser extent and can be found, for example, for:Shoe polish, paints, contact cement, adhesive, lubricants [98];Paints [99];Hygienic cleaning spray cleaners [100];Air care products [101];Disinfectant sprays [102];Antistatic sprays [103];Automobile interior cleaners and deodorizers, antistatic, waterproofing, disinfectant sprays [104].

The exposure scenario should take into account the direction of spraying—into the room (air freshener), horizontally, or vertically onto a surface (cleaning spray, paints) or toward the body (personal care, cosmetic), especially for cosmetic products, contact with the skin is intended and systematically investigated [105]. The use of other products also leads to dermal exposure due to the airborne fraction, e.g., by overspray occurring while spraying onto a surface, especially if the particle size is smaller than 15 µm [106]. The conditions of use like room volume, ventilation rate, and the exposed person(s) should be considered as well.

Spraying activities of workers can have different application patterns, equipment, and behaviour, depending on the sectors and thus have different exposure levels. Information about use characteristics is scattered and thus rather difficult to identify. For spraying of chemicals under REACH, different types of spraying can be identified. However, categorization of spraying in the PROCs, which is described in the ECHA guidance R.12 [107], is very superficial, only distinguishing between industrial (PROC 7) and non-industrial spraying (PROC11). ECETOC TRA for workers uses these PROCs as an initial input parameter. Given that spraying activities at workplace can differ tremendously, using only two PROCs for modelling is a relevant limitation. It would be preferable to have more (sub)-scenarios available within empirical or simple generic models or even within hybrid models. However, Schinkel et al. [108] state that, “For a generic exposure model such as ART, it is not feasible to include very specific determinants such as these for the spraying scenarios. In addition, these determinants are difficult to quantify (based on data used for calibration) and are therefore not included”. For this reason, a better overview is needed on existing sector specific spraying applications, including detailed descriptions and quantitative data on the determinants of relevant spraying activities, reflecting different spray equipment, and use patterns. For example, specific information on use patterns for spraying of diisocyanates—next to other activities with diisocyanates—is described by Rother and Schlüter [109] as an example for a substance of concern that is sprayed regularly in professional and industrial settings.

For spraying of biocidal products, the ECHA collected relevant information in the Guidance on the BPR and related documents [107,110]. Meyer et al. [10] presented for insecticide spraying a manageable summary. Ludwig-Fischer et al. [11] provided a similar paper about wood preservatives (both papers are in German only). A number of BAuA projects dealt with occupational spraying activities for biocidal products and investigated sector-, user-, and use-specific information:Spraying application as such, information on plant protection products in greenhouses, indoor use of biocides, use of disinfectants in animal housing, wood and structure preservation, and antifoulings [4,6];Exposure and protective measures during the application of antifouling paints [111];Exposure patterns and information about use and users during insecticide control of the oak processionary moth, some characteristics of equipment [5];Use of biocidal products (disinfectants and insecticides) as foams in comparison to spraying, different equipment characteristics [16,93].

Further contextual information may be scattered in publications describing the development of spray-specific models [36], the validation of models [6], or studies that measured exposure [112,113,114,115,116].

Different stakeholders collected and presented sector or user specific information about spraying. For example, ConsExpo factsheets include use behaviour information for consumers to some extent. Some of the available use maps also include sector-specific information for spraying activities. For example, the use map of the European Crop Protection Association includes information about the professional and consumer use of plant protection products. The use map of the International Association for Soaps, Detergents, and Maintenance Products includes use information for different relevant product types (e.g., liquid surface cleaners, polishes and wax blends).

The above-mentioned sources demonstrate that worker and consumer spray patterns have similarities but also distinct differences. Some spray equipment such as trigger sprayers are common for both groups. Other types of spraying equipment are only used by professionals, for example, large powered equipment. The same is true for other relevant parameters for exposure assessment such as room dimensions, ventilation regimes, use frequencies, or product amounts. Although a product may be used both by professionals and consumers, the exposure assessments rarely match completely, as the use pattern usually differs in at least frequency or duration of use. Moreover, risk management measures (e.g., containment or personal protective equipment) cannot be considered for consumers [21].

Models, especially mass-balance models, can often be used for both worker and consumer assessment, even if the model itself was developed for only one user group. In case a model is used for assessments of a different user group, it must be ascertained that the selected input values such as use frequency or room volumes are relevant for the user group in question.

In summary, there is a need that the exposure science community should reach agreements on typical default values for specific use scenarios of different spraying activities, user groups, and spraying equipment. Such attempts are, for example, the recommendations of the Ad hoc Working Group on Human Exposure for biocidal products [20] or the publication of sector specific use maps for chemical products [117]. These values and information may change over time when other sources of information become available but also due to actual changes in the work patterns as a result of technological or organisational developments. Thus, they should be updated regularly or as soon as new information is available. In the long term, an independent body would be highly desirable with the aim to evaluate and update default values and information and to develop standard exposure scenarios.

### 6.4. Needs for Modelling from a Regulatory Perspective

From a regulatory point of view, the main need is the development of a platform or database with information on different spraying equipment and techniques and agreed standard parameters for specific spraying scenarios, and harmonization of the terminology.

Spraying activities have the potential for high risks because of a high exposure potential and thus spraying is in the scope of several regulations. Obviously the directly substance related regulations, such as REACH, the plant protection products, and the biocidal products regulations, can profit from such an exchange platform. In addition, less directly substance-related regulations, e.g., construction products regulation (Regulation (EU) No. 305/2011), cosmetics products regulation (Regulation (EU) No. 1223/2009) or medical devices regulation (Regulation (EU) No. 2017/745) could benefit from an improved exchange as they include requirements for risk assessment.

A lot of experience with exposure during spraying has been gained for plant protection. Given the quite numerous numbers of measurements in this sector, these measurements have also been used for the development, calibration, and evaluation of models in different regulatory contexts (e.g., the use of biocidal products). This is appropriate, as exposure does not necessarily depends on the regulatory context or the class of the substance. Of course, exposure scenarios are sometimes different (e.g., indoor or outdoor, used volume) and determine whether an extrapolation from one regulatory context (plant protection) to another regulatory context (biocidal products) is appropriate. As long as substance/product properties, spraying equipment, and other relevant determinants are similar (or similar enough), exposure assessment methodologies in the different regulations can learn from each other. With regard to dermal exposure, it should be noted that measurement data collected for plant protection products may include significant contact with treated plants and this limits transferability to other regulatory areas.

Generic models (tier-1 models, screening tools) provide generic exposure estimations that are meant to be very conservative for most spraying situations. For the assessment of a wider range of scenarios, models with a more mechanistic approach (e.g., ConsExpo or SprayExpo) and other simulation approaches offer promising alternatives. Pre-defined sector-specific parameters, for example, data sheets or predefined selection lists, may significantly improve user acceptance of these models. This approach is followed, for example, by ConsExpo, which is supported by factsheets describing consumer applications. In general, model-supporting factsheets are a useful tool to support users and provide relevant contextual information, for example, on user behaviour or exposure situations. Another option are model integrated standard scenarios, as for SprayExpo. However, these approaches are scarce and model-specific. In order to promote the implementation of such features, more comprehensive information and documentation on the parameters of spraying equipment would be required. Next to these scarce examples of standard scenarios and factsheets, only very little is harmonized and standardized. This accounts especially for semi-automated and automated spray processes that are of interest for workplace assessments. Currently these uses can be assessed with ART and possibly with ECETOC-TRA for workers. However, this would either be an intricate or a very generic assessment and the evaluation status of the models with respect to such applications is scarce.

This brings us back to the idea and need for a (exchange) platform. It could be used additionally to simplify or reduce the number of input parameters required by the models, e.g., by combining interdependent parameters (e.g., treated area increases with room size; applied amount increases with treated area).

To avoid any misinterpretations between different stakeholders, different spraying application techniques or spraying equipment should be named consistently within different regulations, in the exposure models and for products on the market. For example, equipment such as a pump sprayer or trigger sprayer have to be specified. A starting point could be descriptions such as those presented in the emission scenario document for insecticides, acaricides, and products to control other arthropods for household and professional uses [118,119,120,121,122,123,124,125,126,127,128,129,130,131,132,133,134,135,136,137,138,139,140,141,142,143,144,145].

## 7. Conclusions

Various models exist (partly implemented in tools) that are used to estimate human exposure to spray mist. These models have been briefly described and summarized in this paper and information on their performance (e.g., applicability domain, accuracy or uncertainty) has been given where available. The evaluation of the models and the assignment to an applicability domain depends on the availability of reliable and well-documented exposure measurements for spray applications. However, such studies providing the required measurement data are scarce and the available studies often lack important contextual information that would be required for an accurate and reliable comparison of the measurements with the modelling results. In this paper, we have, therefore, placed emphasis on outlining the information required for model development and evaluation.

We identified the following needs for improving the modelling of exposure by spraying activities:Extension of mechanistic model approaches to cover post-application phases as well as spraying of (semi-)volatile substances, i.e., combined exposure to spray mist, evaporation from droplets, and evaporation from treated surfaces;Further research and practical implementation of concepts to consider the type and amount of the variable source strength for different spray equipment in the modelling;Development and evaluation of models for exposure assessment of foaming activities and dermal exposure assessment of spray activities;Comprehensive evaluation of empirical, hybrid, and mechanistic models with a focus specifically on different spray scenarios and equipment;A better documentation and guidance of the models, e.g. description of which spray scenarios and spray equipment are covered by the models and evaluation studies;Development of a database with agreed default parameters for specific spraying scenarios and source parameters for different spraying devices and techniques;Harmonization of terminology, spray input parameters and appropriate grouping of spray exposure situations among the models;A platform with harmonized information about spraying activities and appropriate exposure models to be used under different regulations.

In the context of an ISES Europe workshop (Workshop on “Theoretical Background of Occupational Exposure Models”, https://ises-europe.org/exposure-platform/data-and-information-sharing) (accessed on 9 June 2021), ECHA presented a few ideas on the generic requirements of modelling in regulatory contexts. Our findings are in line with these requirements. For example, ECHA identified the need to establish a platform for developing a common framework that supports assessors (choosing the most adequate method (tool(s) or measured data) for the substance and use-situation to be assessed), users of chemical products (communication of safe use advice), and regulators (improve the acceptance).

In summary, a number of improvements have been identified that would greatly increase user and regulatory acceptability and would help to integrate mechanistic spray models into practicable and effective, risk-based regulatory strategies.

## Figures and Tables

**Figure 1 ijerph-18-07737-f001:**
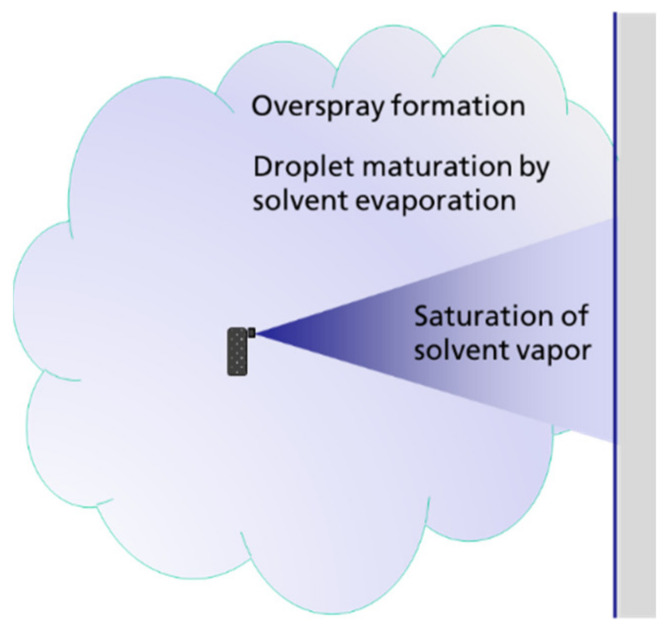
Illustration of the aerosol formation during spraying activities. Non-deposited droplets constitute the overspray. Overspray droplets mature due to solvent evaporation and shrink to a final size depending on the concentration of the non-volatile component in the formulation. Inside the spray cone, solvent vapour may reach saturation due to the large surface area of the dispersed liquid.

**Table 1 ijerph-18-07737-t001:** Tools generally suited to estimate human exposure to spray mist; a short description of the underlying model structure of each of these and few other tools as well as the input parameters and the output data are given in Appendix A. REACH stands for Registration, Evaluation, Authorisation and Restriction of Chemicals.

Tool/Model	Scope	Target	Spray Model	Route	Reference
AISE REACT	Household products (related to PC 35 & PC 3 under REACH).	Consumer	Simple mechanistic model for deterministic exposure estimates	inhalation, dermal	[38,39]
AOEM	Plant protection products (Outdoor and greenhouse)	Occupational bystander	Empirical approach by using categories for stochastic exposure estimates	inhalation, dermal (body, hands, head)	[9,27,40]
ART	Various work place situations under REACH	occupational	Conceptual framework based on scores and modifying factors calibrated with measured data	Inhalation	[31,32]
BROWSE	Plant protection products	Occupational bystander	Mechanistic approach calibrated with measured data	Inhalation, dermal	[41,42,43]
CEM	Several models for consumer product applications	Consumer	Mechanistic model for deterministic exposure estimates	inhalation	[44]
ConsExpo	Several models for consumer product applications	consumer, (occupational)	Simple and sophisticated mechanistic models for deterministic and stochastic exposure estimates	inhalation, dermal	[23,45]
dART	Various work place situations Adaptation of ART for dermal exposure	occupational	Stochastic hybrid model including BEAT and BROWSE findings	dermal	[3,46,47]
ECETOC TRA consumer EGRET	Household and DIY products related to PCs under REACH	consumer	Simple mechanistic model for deterministic exposure estimates	inhalation, dermal	[48,49,50,51,52,53,54]
ECETOC TRA worker	Various work place situations under REACH Screening tool	occupational	Empirical approach for deterministic exposure estimates (75th percentiles) for industrial and professional spraying	inhalation, dermal	[48,49,50,51,52,53]
MEASE	Various work place situations under REACH with focus on metal processing	occupational	See ECETOC TRA worker	inhalation, dermal	[55]
RISKOFDERM	Various work place situations for dermal exposure	occupational	Empirical approach for stochastic exposure estimates	dermal (body, hands)	[56,57]
SprayExpo	Manual spraying activities for non-volatiles	occupational, (consumer)	Sophisticated mechanistic model for deterministic exposure estimates on a higher tier level	inhalation, dermal	[4,6]
Stoffenmanager	Various work place situations in the framework of OSH, national legislations, and REACH	occupational	Conceptual framework calibrated with measured data for stochastic exposure estimates	Inhalation (+ qualitative dermal control banding module)	[58,59,60]
TNsG spraying models	Biocidal products	occupational, consumer	Empirical approach with several models for stochastic estimates	dermal (body, hands), (inhalation not all models)	[20,61,62]

## Data Availability

Data is contained within the article or Appendix A.

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
