# Peer review of "Modelling Exposure by Spraying Activities—Status and Future Needs"

_ijerph, 2021, doi:10.3390/ijerph18157737_

Round 1
Reviewer 1 Report
I read the manuscript once then went back for a detailed review. There are many issues with the English. I highlighted a number of them but stopped reading because I do not want to review English.
There are other many other issues such as the methods and materials section that does not actually contain methods and materials information. Organizationally there should be a section where "future needs" are clearly presented (to go along with the title). I would be happy to take another look when the English and organization are improved.

Author Response
We agree that the standard headings of research articles are not suitable for such a review. For this reason the headings were adapted. The chapter 2 describe now the approach used for the literature review (Details are given in the supplementary material). The workshop has been used to get information from experts which are not mentioned in any literature source.
Future needs were primarily mentioned in the discussion section. This is now the heading “identified needs” and the section gives a detailed discussion of the identified needs. A summary of the needs are given in the conclusion.
The yellow parts in the paper have been revised, and based on this comments also the other parts were subject of an English editing, and we tried to be more clear in the terms used.
Reviewer 2 Report
This article concerns an evaluation about occupational exposure to spraying activities. The text appears to be unbalanced in its underlying structure. In my opinion, the paper needs a thorough revision and re-editing. Some observations (not exhaustive):
- The paper’s aims should be better clarify.
- The methodology by which the review was carried out is not adequately described and the type of revision is unclear. It would seem a narrative review, but even in this case all the minimum data reporting and selection process are required. For this reason it is necessary to specify the process fo identifying the literature search (e.g. years considered, language, publication status, study design and coverage database).
- Paragraph 2.1 seems redundant, not useful for discussion. The workshop is already mentioned in the abstract, it is sufficient to refer to it in the final part of the introduction.
- Please, correct the page numbering.
- Section 3.1 seems to belong more to the introduction than to the results.
The article should be more summarized and optimized, as it is difficult to read in some parts
Author Response
Thanks for your review and comments which were very helpful.
We restructured the paper as follows:
We specified in more detail the methodology used for the review. For example we repeated the literature search in the databases. Described that in the section 2, now named literature review. The details of the literature search is given in the supplementary material including a PRISMA flow diagram. We hope this is in agreement with your expectations.
The other headings were adapted as method, result and discussion does not work for this review.
As you suggested we shifted the section 3.1 to the introduction, and revised the introduction accordingly.
We rephrased the objectives of this review, and thus clarified the aim of the paper at the end of the introduction
Paragraph 2.1: Yes the workshop was already mentioned in the abstract, and at different parts of the review, i.e. methods and needs. We think the workshop/symposium was an important source of information from experts on the knowledge and needs of the different models. For this reason we assigned the workshop to the method section. We added some sentences to describe the literature search and why the workshop was used to getting information from experts. A detailed description of the literature search is now presented in a PRIMSA flow diagram as part of the supplementary material. In addition, we shortened the information in the abstract and the needs section to be not redundant.
Page numbering has been corrected. Sorry for that.
Finally, we edited all the sections, and tried to give a better structure of the content.
For your information, in the chapter of the description of the models it was not easy to find a clear structure, for example differentiate between consumer or worker models, or clearly between mechanistic and empirical model. For this reason we structured from simple empirical to more complex empirical to the hybrid models which contain empirical as well as simple mechanistic parts. After that we described the simple mechanistic models and the complex mechanistic models. In our view this was the best way to describe the models.
This manuscript is a resubmission of an earlier submission. The following is a list of the peer review reports and author responses from that submission.